# Characterization of Five Collagenous Biomaterials by SEM Observations, TG-DTA, Collagenase Dissolution Tests and Subcutaneous Implantation Tests

**DOI:** 10.3390/ma15031155

**Published:** 2022-02-02

**Authors:** Miki Hoshi, Tomofumi Sawada, Wataru Hatakeyama, Masayuki Taira, Yuki Hachinohe, Kyoko Takafuji, Hidemichi Kihara, Shinji Takemoto, Hisatomo Kondo

**Affiliations:** 1Department of Prosthodontics and Oral Implantology, School of Dentistry, Iwate Medical University, 19-1 Uchimaru, Morioka 020-8505, Iwate, Japan; hoshmiki@iwate-med.ac.jp (M.H.); whatake@iwate-med.ac.jp (W.H.); ykhcnh@iwate-med.ac.jp (Y.H.); takafuji@iwate-med.ac.jp (K.T.); hkihara@iwate-med.ac.jp (H.K.); hkondo@iwate-med.ac.jp (H.K.); 2Department of Biomedical Engineering, Iwate Medical University, 1-1-1 Idaidori, Yahaba-cho 028-3694, Iwate, Japan; sawada@iwate-med.ac.jp (T.S.); takemoto@iwate-med.ac.jp (S.T.)

**Keywords:** collagenous biomaterial, guided bone regeneration membrane, scanning electron microscopy, thermogravimetry-differential thermal analysis, collagenase dissolution test, subcutaneous implantation test, cross-linking of collagen fibrils

## Abstract

Collagenous biomaterials that are clinically applied in dentistry have dermis-type and membrane-type, both of which are materials for promoting bone and soft tissue formation. The properties of materials supplied with different types could affect their biodegradation periods. The purpose of this study was to characterize five of these products by four different methods: scanning electron microscopy (SEM) observation, thermogravimetry-differential thermal analysis (TG-DTA), 0.01 wt% collagenase dissolution test, and subcutaneous implantation test in vivo. SEM micrographs revealed that both dermis and membranous materials were fibrous and porous. The membranous materials had higher specific derivative thermal gravimetry (DTG) peak temperatures in TG-DTA at around 320 °C, longer collagenase dissolution time ranging from about 300 to 500 min, and more longevity in mice exceeding 9 weeks than the dermis materials. There existed a correlation between the peak temperature in TG-DTA and the collagenase dissolution time. It was considered that higher cross-link degree among collagen fibrils of the membrane-type collagenous materials might account for these phenomena. The experimental protocol and numerical results obtained could be helpful for selection and future development of fibrous collagenous biomaterials in clinical use.

## 1. Introduction

Collagen is the oldest and most plentiful extracellular matrix protein that has found many uses in food, cosmetic, pharmaceutical, and biomedical industries [1,2]. Especially, collagen has been widely employed in the medical fields including artificial skin in dermatology, heart valves and vessel replacement in cardiovascular surgery, hernia repair, adhesion barriers and tissue adhesives in general surgery, nerve conduits and repair in neurosurgery, vitreous replacement and retinal reattachment in ophthalmology, bone repair and cartilage repair in orthopedics, ureter replacement, renal repair and dialysis membrane in urology [3,4,5,6,7,8,9] and dental fields including epidermal regeneration, periodontal attachment and alveolar ridge augmentation [10]. Nowadays, a new era for tissue engineering and biotechnology using collagenous biomaterials has emerged, which includes scaffold materials with growth factors and stem cells for tissue regeneration and three-dimensional (3D) printing technology for artificial organs [11,12]. Hence, the application of collagenous biomaterials has currently been categorized into food and beverages, wound healing, healthcare, cosmetics and skin, drug delivery, bone disease, dental applications, tissue engineering, and 3D printing [1].

In dental applications, collagenous biomaterials are widely used as artificial dermis, filling materials for tooth extraction socket and membranes for guided tissue regeneration (GTR) and guided bone regeneration (GBR) [13,14,15,16,17,18]. That is, both types of collagenous biomaterials contribute to promoting bone and soft tissue formation. The properties of materials supplied with different types could affect their biodegradation periods. As for GBR membranes, polylactide based polymer as well as collagen was used, and attention was paid to these two biomaterials [18]. Gel and fiber scaffolds consisting of collagen have been applied for bone regeneration in clinical trials [1,2]. In vivo, the sponge-like dermis materials stay for at least 4 weeks in humans, while the membranous materials remain for about 6 to 24 weeks [18,19,20,21,22]. These collagenous biomaterials are frequently utilized in clinical use, however, the morphology, structure, level of cross-linking and in vitro/in vivo durability of them have not been well simultaneously scrutinized yet [23].

The collagen molecule single-handedly is not stable. In nature, it is organized into a triple helix structure, also termed a collagen fibril. Many fibrils are then assembled with a covalent cross-linked bond to get a collagen fiber. Thus, there are various types of collagens. Collagen from type I to type IV is common in a human body. Type I collagen is the most applied natural biomaterial in dental applications as the main constituent of the extracellular matrix, it is biocompatible and biodegradable [18]. Collagen may be separated from animal tissue. To reduce the antigenicity of collagen, the terminal telopeptides of collagen are eliminated enzymatically to acquire atelocollagen [24]. To be utilized in dental practice, extracted type I collagen and atelocollagen often need to be crosslinked to raise mechanical strength and enzymatic resistance [18]. There are chemical and physical methods accessible to increase collagen properties [18]. Chemical crosslinking is most achieved by glutaraldehyde, carbodiimide/hydroxysuccinimide, diisocyanates, or oxidized sugar [18,25,26]. For physical cross-linking, dehydrothermal treatment and ultraviolet radiation are frequently applied [26,27].

Moreover, collagenous products were usually manufactured from type I collagen fibers derived from porcine skin, bovine skin, or bovine tendon in dental applications. Briefly, atelocollagen or purified type I collagen fibers were dispersed in an acidic media (pH 2.5), homogenized, filtered, reconstituted (pH 7), dehydrated, compressed, freeze dried, cross-linked (or not done), sized, packaged, and sterilized [28]. As a result of the complicated manufacturing process, many variations take place in physical, chemical, and biological properties of collagenous biomaterials [1,2]. The manufacturing process from animal skins and tendons usually consists of decellularization, acid solubilization, (not often, alkaline treatment), salt precipitate, filtration, dialysis, (some-times, gel column separation, and electrophoresis) with repeated steps, and all sequential steps could be sources in variations of properties of collagen [1,2,28]. The manufacturing process and alteration in properties of collagen is highly dependent on the manufacturers. Additional cross-linking also causes further variation. Thus, detailing the manufacturing process is beyond the scope of this study because it is not disclosed. However, property analysis of these materials seemed quite useful to obtain selection criteria for clinical use and standard of future development for them [29]. It will also contribute to evidence-based explanation to patients and the clinical evaluation of the collagenous biomaterials.

Characterization of collagenous biomaterials has been historically performed by physical, chemical and biological methods [30]. Briefly, it is divided into four major areas such as (i) structural detail mainly focused on molecular mass, purity, helical content, and bulk thermal properties, (ii) chemical feature mainly focused on surface elemental analysis, hydrophobicity and compositions, (iii) morphological features at different length scales and (iv) physical and biological features. Regarding (i), sodium dodecyl sulfate-polyacrylamide gel electrophoresis (SDS-PAGES) is most used to assess collagenous biomaterial purity and breakdown. Subsequent Western blots can be used to assess the specificity of collagen type using monoclonal antibodies [31]. Matrix-assisted laser desorption ionization time-of-flight mass spectroscopy is further used to provide molecular mass data on collagen as well as the identification of telopeptides and other potential contaminants in collagen [32]. Circular dichroism (CD) utilizes the differential absorption of circular polarized light to assess the helical content of collagen [33]. Differential scanning calorimetry (DSC) thermal analysis reveals transitions in the structural state (e.g., dissociation temperature of helix structure) of collagen-based materials, reflecting degree of crosslinking [34]. Next, regarding to (ii), the elemental composition of collagen biomaterials can be determined by X-ray photoelectron spectroscopy (XPS) [35]. The hydrophobic character of collagenous surfaces is often characterized using water contact angle measurements [36]. Hydroxyproline assay is customarily used to determine collagen content although metabolic labeling with radioactive amino acids, high-performance liquid chromatography and colorimetric assays have been alternatively proposed [30]. Ninhydrin assay is utilized to quantify the amount of free amino acids. Ninhydrin reacts with the primary free amino groups of the protein and a color change, from yellow to purple, occurs [37]. 2,4,6-trinitrobenzene sulfonic acid assay is also used as a means to quantify free amino groups. The concentration of N-trinitrophenyl protein derivatives is measured by molecular absorption spectroscopy at 345 nm. The degree of cross-linking of collagenous biomaterials can be calculated using free amine quantities [38]. In vitro enzymatic degradation of collagenous biomaterials by matrix metalloproteinases, usually MMP-1 (collagenase 1), allows investigation of the stability of type 1 collagenous biomaterials [32,39]. Regarding (iii), surface imaging tools of collagenous biomaterials include atomic force microscopy (AFM), scanning electron microscopy (SEM), environmental scanning electron microscopy (ESEM), and light microscopy, corresponding from the large to small magnifications in this order [30]. Finally, regarding (iv), a mechanical test, the tensile strength test, has been conducted for determination of bulk strength of collagenous biomaterials [19]. Differential thermal analysis (DTA) of collagenous biomaterials has also been carried out as another physical test [40,41]. Animal tests such as subcutaneous implantation of collagenous biomaterials and in vivo implantation test of the construct consisting of collagen and cells have also been performed [42,43]. Mechanobiology has been an important considering matter of collagenous biomaterials, especially for tendon repair using collagen-based scaffold, with which tendon stem cells can be simultaneously applied to the repairing area, susceptible to high mechanical stress [44].

Therefore, with reference to these findings, the purpose of this study was undertaken to conduct SEM observations to provide vivid morphological images, thermogravimetry-differential thermal analysis (TG-DTA) to unveil structural state and level of cross-linking of collagen, collagenase dissolution tests to represent in vitro chemical durability, and subcutaneous implantation tests to show the direct in vivo longevity using five commercially available collagenous products in dental practice.

## 2. Materials and Methods

### 2.1. Materials

Five commercial collagenous biomaterials used in this study are shown in Table 1. One artificial dermis (Terudermis: TD) and one filling material (Teruplug: TP) for tooth extraction sockets are produced by the same manufacturer (Olympus Terumo Biomaterials Corp., Tokyo, Japan). As for animal origins of collagen, these artificial dermis-type materials (TD and TP) were made from bovine skin.

The other three products are clinically utilized as GTR/GBR membranes; Koken Tissue Guide (KT; Koken CO., Ltd, Tokyo, Japan), Biomend (BM; Zimmer Biomat Dental G.K, Warsaw, IN, USA) and Bio-gide (BG; Geistlich Pharma AG, Wolhusen, Switzerland). One GTR membrane material (KT) was manufactured from the mixture of bovine skin (90 wt%) and tendon (10 wt%). One GTR/GBR membrane material (BM) was formed from bovine tendon. One GTR/GBM double-layered membrane material (BG) was created from porcine skin.

### 2.2. Methods

#### 2.2.1. SEM Observations

Outer and cross-sectional surfaces of each sample (*n* = 1) with the dimension of about 4 × 4 × 1.5 mm were examined by using an SEM (SU8010, Hitachi High-Tech Corp., Tokyo, Japan) at 15 kV after plasma-coating with OsO_4_. The cross-sectional surface of the sample was exposed by cutting with scissors. The pore size and thickness of the layered structure of the sample were measured using the ImageJ software (the National Institute of Health, MD, USA). The average pore and layer sizes were calculated from different locations per one sample (*n* = 20).

#### 2.2.2. TG-DTA

TG-DTA was performed on each sample (*n* = 1) weighing from 3.19 mg (KT), 6.36 mg (TD), 6.49 mg (BG), 6.52 mg (BM), and 6.80 mg (TP), employing specialized equipment (TG/DTA 6300, Hitachi High-Tech). The experimental conditions were as follows: atmospheric gas; Nitrogen, gas flow rate; 200 mL/min, temperature range; room temperature to 1000 °C, heating rate; 10 °C/min, sample holder; open platinum crucible, reference; alumina powder (6.75 mg). First derivative of thermal gravimetry (TG) curve was termed as derivative TG (DTG) curve. TG-DTA informs mass (weight) change in TG and thermal properties and phase changes in DTA of samples upon heating [40,41].

#### 2.2.3. Collagenase Dissolution Tests

Each sample (1.5 mg) (*n* = 6) was dissolved in 0.01 wt% collagenase (S-1, Nitta Gelatin Inc., Osaka, Japan) solution diluted in distilled water (1 mL) in a 1.5 mL microtube, and had been placed in a constant temperature bath, kept at 37 °C. The dissolution condition was visually inspected every 6 h, and the time to totally disappear (min) was recorded.

#### 2.2.4. Subcutaneous Implantation Tests

Subcutaneous implantation tests were performed using five 10-week-old male Jcl:ICR mice (Clea Japan Inc., Tokyo, Japan) in accordance with the guidelines for the care and use of laboratory animals and was approved by the Institutional Ethics Committee of Iwate Medical University on 19 Mar 2021 (approval number: 02-035). The number of mice used was minimized so that the principles of the 3Rs (Replacement, Reduction and Refinement) for animal welfare could be protected [45]. After depilating, the mice were anesthetized with mixed gas of isoflurane and oxygen, and five samples in the size of about 4 × 4 × 1.5 mm were subcutaneously implanted in the back tissue of a mouse with scissors, scalpel, and tweezers. The wounds were closed with nylon sutures. The mice were then fixed with free water and diet. The mice were sacrificed at 1, 2, 3, 4 and 9 weeks (*n* = 1) after implantation, and the tissues surrounding the specimens were exposed and visually assessed. In this study, female mice were not used to eliminate the influence of wound healing by hormones [46].

#### 2.2.5. Statistical Analysis

The data of the collagenase dissolution tests were statistically analyzed using the software package BellCurve for excel (Social Survey Research Information Co., Ltd., Tokyo, Japan) with Welch’s one-way analysis of variance (ANOVA) followed by Bonferroni’s multiple comparison tests at a level of significance of α = 0.05.

## 3. Results

### 3.1. SEM Observations

The micrographs of the outer and cross-sectional surfaces of each sample are shown in Figure 1. For two dermis-type materials, the fibrous and porous structure was observed in both TD (Figure 1a,b) and TP (Figure 1c,d). The pore sizes on the outer surface in TD and TP were 99.9 ± 69.6 and 42.3 ± 34.8 µm, whereas those on the cross-sectional surface were 139.3 ± 77.0 and 150.4 ± 88.4 µm, respectively.

For three membranous materials, in KT, the outer surface was fibrous and dense, but the cross-section surface consisted of the layered plate structure with ample porous rooms (Figure 1e,f). The penetration of the porosity from the surface to the interior was not clear. The pore sizes on the outer surface in KT were 212.7 ± 115.2 µm. The wall thickness, horizontal and vertical sizes of porous rooms on the cross-sectional surface in KT were 5.4 ± 2.2, 100.9 ± 42.9, and 13.0 ± 4,6 µm, respectively.

The most outer surface was fibrous and densely packed in BM, while the cross-section surface was made up of the layered plates with pore spaces (Figure 1g,h). The path through from the surface to the interior did not exist at this magnification. The pore sizes on the outer surface in BM were 47.8 ± 20.6 µm. The wall thickness, horizontal and vertical sizes of porous rooms on the cross-sectional surface in KT were 5.0 ± 2.1, 74.1 ± 38.0, and 11.9 ± 3.2 µm, respectively.

In BG, two layered structures were cross-sectionally observed (Figure 1i,j). The outer-most surfaces were fibrous and highly-compacted in both top (Figure 1i) and bottom (Figure 1j) sides. Apparent pores were not found in both sides on outer surfaces. The cross-sections of the top and bottom sides were very smooth but dense at the epithelium side and porous and rough at the bone side, respectively (Figure 1k). The cross-sectional inter-connection was not found, either. The compacted fibrous structure was also observed without pores on the top side. The wall thickness, horizontal and vertical sizes of porous rooms of the bottom side on the cross-sectional surface in KT were 6.9 ± 4.1, 90.4 ± 69.7, and 24.7 ± 8.4 µm, respectively.

### 3.2. TG-DTA

TG-DTA of each sample are shown in Figure 2. The DTG peaks and the concurrent weight loss (wt%) of each sample are also summarized in Table 2. By proportionating respective peak temperature with corresponding weight loss of collagen samples, the specific DTG peak temperature was determined from two peak temperatures 1 and 2 at 220 °C and 320 °C for each sample (especially for TD, TP, and BG).

In the TG-DTA of TD and TP as dermis-type materials, respective water evaporation and endothermic peak were seen at 65.7 °C and 65.2 °C with slight weight loss on TG leading to a small DTG peak (Figure 2a,b). In TD (Figure 2a), at 215.4 °C, the big weight loss was noticed on the TG curve accompanied bya large DTG peak and minute exothermic DTA peak. Moreover, at 322.5 °C, the gradual weight loss on TG, medium and broad DTG peak and a small endothermic DTA peak were observed in TD. In TP (Figure 2b), at 206.3 °C, the gradual weight loss was noticed on the TG curve accompanied by a medium DTG peak and small and broad endothermic peak on DTA. Furthermore, at 326.5 °C, a medium DTG peak due to the gradual weight loss on TG was observed with an endothermic peak on DTA in TP.

Among three membranous materials (Figure 2c–e), water evaporation and an endothermic peak was seen on DTA with slight weight loss on TG leading to a small DTG peak at 58.0 °C in KT, 72.2 °C in BM, and 63.2 °C in BG, respectively. In addition, the weight loss on TG was gradually observed with the medium DTG peak along with the medium broad exothermic peak on DTA at 326.1 °C in KT, 315.8 °C in BM, and 328.9 °C in BG, respectively.

### 3.3. Collagenase Dissolution Tests

The dissolution time of each sample is presented in Figure 3. The dissolution time of two dermis-type materials (TD and TP) was short at around 40 min, while those of three membranous materials (KT, BM and BG) were much longer, ranging from about 300 to 500 min. The latter three showed statistically significantly longer dissolution time than the former two (*p* < 0.01).

Among three membranous materials, BM showed shorter dissolution time than KT and BG (*p* < 0.01), while there was no statistically difference between KT and BG.

### 3.4. Subcutaneous Implantations Tests

The postoperative photograph of each sample after the subcutaneous implantation is presented in Figure 4. The dermis-type materials (TD and TP) remained in mouse back subcutaneous tissues up to three weeks, but disappeared four weeks after implantation (Figure 4a). On the other hand, the membranous materials (KT, BM and BG) remained still nine weeks after implantation in vivo (Figure 4b).

## 4. Discussion

The morphological information was vital and pre-requisite for understanding of functioning and biodegradation of collagenous biomaterials. Besides the images, two important in vitro properties and one in vivo property were prudently selected and evaluated in this study.

SEM provides morphological features of collagenous biomaterials which considerably affect clinical performance, as explained below. By SEM observations (Figure 1), the dermis-type materials (TD and TP) were fibroblastic and porous with pores highly inter-connected. Fibroblasts might easily infiltrate, and replace these materials with natural connective tissues during wound healing [47]. On the contrary, outer-most surfaces of two membranous materials (KT and BM) were smooth and dense which might hinder the invasion (i.e., downgrowth) of epithelium tissues at the top, but allow regeneration of bone, periodontal ligament, and cementum at the bottom [48]. BG, one membranous material, had the bilayered structure. The cross-section of the top side was denser in the epithelial side to hinder epithelial tissue downgrowth, while that of the bottom side was loose and rough in the bone side to accelerate bone formation (Figure 1k). Inter-connection from the outer surface into the interior of membrane materials was not evident. Plasma-coating with OsO_4_ might uncover the small connecting holes at the size of about 10 µm [49], which might permit flow of oxygen and nutrition.

TG-DTA appeared to be effective in evaluating the degree of cross-linking of collagenous biomaterials. It has not been utilized for this purpose to date, although DSC thermal analysis was more popular [28]. It became evident from the results obtained that the dermis-type materials (TD and TP) were less cross-linked with lower specific DTG peak temperature, less collagenase dissolution time and shorter in vivo remaining period in subcutaneous tissue, compared with the membranous materials (KT, BM and BG). The rational for these phenomena is considered here from the material characteristics. According to the manufacturer, artificial dermis (TD) and filling material for tooth extraction socket (TP) were produced by the similar manufacturing process, starting from purified atelocollagen in acid [50]. The major part (90 wt%) of reconstituted atelocollagen (i.e., fibrotic atelocollagen) was mixed with the minor part (10 wt%) of heat-denatured atelocollagen, followed by dehydrothermal cross-link. Due to the lack of strong chemical cross-link, these artificial dermis-type materials became more feasible to biodegradation in vivo. On the other hand, the membranous materials (KT, BM and BG) might be prepared by the method, described in the introduction section, employing various extracted type I collagen fibers, as start material, which lost terminal telopeptides by enzyme treatment but still maintained abundant naturally formed (lysyl oxidase-mediated) cross-links among collagen fibrils [24,51]. One sample used in this study (KT) was further chemically cross-linked with hexamethylene diisocyanate [52]. Another sample (BM) might be chemically cross-linked with formaldehyde [22]. Therefore, the biodegradation of these membranous materials would be much delayed, proportionally to the level of cross-linking, with respect to the absorption period of the dermis-type materials. One membranous material (BG) was made from native porcine skin with the bilayered structure, the bone side of which might in part contain quick-disappearing fibrotic atelocollagen but the epithelium side of which might contain more slow-deteriorating (cross-linked) type I collagen fibers [22].

In TG-DTA (Figure 2), two characteristic DTG peaks were found. The change of collagen during heating associated with these peaks is schematically drown (Figure 5). It is known that re-constituted collagen (fibrotic atelocollagen) with helix configuration is transformed to random single collagen at 220 °C (peak temperature 1) [41], while lots of intrinsic water is released with large TG weight loss (Figure 5a). This TG drop was a proof of the existence of fibrotic atelocollagen in collagen products. Cross-linked collagen (collagen fibers) was made up of varied molecular weights, and was thermally decomposed in the long-range temperature. It could be characterized by the DTG peak at around 320 °C (peak temperature 2) as a result of bulk degradation of dried cross-linked collagen fibrils [41]. The thermal dissociation at 320 °C is shown schematically in Figure 5b. It is no wonder to think that the increase in the cross-linking of collagen fibrils brings about the higher DTG peak temperature 2. In this study, two DTG peak temperatures (1 and 2) were combined into a specific DTG peak temperature, considering the contribution of respective TG weight loss. The evaluation of the cross-linking of collagenous biomaterials by thermal analysis has not been popular, but appears promising for future studies. In industry, the experimental protocol was proposed [53]. In other research, a different type thermal analysis, DSC, has been employed up to about 300 °C, at which the onset of the dissociation of fibrotic atelocollagen could be well monitored [28,54].

It can be pointed out that the collagenase dissolution test directly indicates the degree of cross-linking of collagen biomaterials [18]. The dissolution tests in this study were performed by the visual inspection because the collagenase quantity measuring by the collagenase kit assay did not determine the level of the cross-linking of collagenous biomaterials. In addition, the colorimetric assay using 2,4,6-trinitrobenzene sulfonic acid assay to quantify free amino groups [38] was not determined by the level of the cross-linking of collagenous biomaterials, depending on the materials. In the results of the preliminary experiment, the high level of the cross-linking such as KT, BM and BG could not be dissolved within a week at 37 °C by sodium bicarbonate solution (pH 9) and acid (pH 3) with pepsin solution. The in vivo biodegradation of the collagenous biomaterials mostly takes place by endogenous collagenases into carbon dioxide and water [24]. The cross-linking of collagen fibrils renders collagenous biomaterials more resistant to deterioration by collagenase, degree-dependently. The higher the cross-linking of collagen, the longer the longevity of collagenous biomaterials in vivo. Porous materials are more prone to deterioration by collagenase due to increased contact between substrate and collagenase enzyme in solution. The porous structures of the dermis-type materials were clarified by SEM observations (Figure 1a–d). In these regards, the dermis-type material with lower degree of cross-linking and higher surface area is more digested than the plain layered membranous materials (Figure 1e–k). Indeed, the dermis materials (TD and TP) possessed much shorter dissolution time of about 30 to 50 min than the membranous materials (KT, BM and BG) with dissolution time of about 300 to 500 min (*p* < 0.01) (Figure 3). The membranous materials were more cross-linked, but the difference in the collagenase dissolution time was noticed among the materials because of different level of the cross-linking in collagen fibrils. Among the membranous materials, the non-cross-linked material (BG) showed significantly more longevity in the dissolution test than the cross-linked materials (KT and BM). The obtained results in this in vitro study seemed to be consistent with the previous study [10], that the resorption time of the non-cross-linked material was longer than the cross-linked materials for clinical use. Therefore, the dermis-type materials appear to be suitable for wound healing of soft tissues, while the membrane materials could be applied to bone augmentation from the standpoint of the material longevity in vivo and regenerative capability of collagen for both soft and hard tissues [1,2].

The correlation of the specific DTG peak temperature and collagenase dissolution time of five products is considered next. Figure 6 displays the correlation between the specific DTG peak temperature and collagenase dissolution time of five collagenous biomaterials. There existed a relatively strong positive logarithmic correlation (r^2^ = 0.7057) between two properties considered. It could be pointed out that the specific DTG peak temperature might be another gauge, indicating the level of the cross-linking in collagenous biomaterials.

The subcutaneous implantation test also directly provided the level of the cross-linking of collagenous biomaterials. The dermis-type materials (TD and TP) disappeared in the subcutaneous tissues 4 weeks after implantation due to low degree of the cross-linking (Figure 4a). The membranous materials (KT, BM and BG) remained in subcutaneous tissues 9 weeks after implantation (Figure 4b). It was reported that one membranous material (BG) endured in subcutaneous tissues up to 20 weeks after implantation [55]. However, the results of subcutaneous implantation tests were consistent with in vitro experimental outcomes by TG-DTA and collagenase dissolution tests in this study. Considering clinical conditions, dental collagenous biomaterials are inserted to the site of bleeding, clot formation, vascularization, and granulation formation, associated with wound healing and subsequent bone formation. Under such circumstances, collagenous biomaterials might be more extensively metabolized with shorter life-span, while matrix metalloproteinases including collagenase are abundantly secreted from connective tissues during wound healing, and body-defense cells produce much invasive inflammatory cytokines, acid and free radicals [56].

In dentistry, longer in vivo remaining period is expected for collagenous biomaterials, especially, for GBR membranes. Materialistic research is highly anticipated [29]. The in vitro data revealed in this study seemed to provide numerical scales of collagenous biomaterials for selection in dental practice. The property of the collagenous biomaterials can be changed (modified). The dermis-type collagenous biomaterials (TD and TP) can be more collagenase-resistant by chemical and physical cross-linking. Especially, glutaraldehyde is a strong cross-linker agent [28]. The dermis-type and GTR/GBR membrane collagenous materials appeared similar, but were selected and grouped differentially in this study because of the varied biodegradation periods and clinical use [1,2], and they are not interchangeably employed. Considering ease in the cross-link technology, the dermis-type materials could be cross-linked by the chemical agents so that the durability against collagenase was increased. On the other hand, the GTR/GBR membrane materials could not be converted to the collagen sponges with the lower cross-link level similar to the dermis-type materials by the simple chemical and physical methods. Decrosslinking of the chemically cross-linked collagen such as the GTR/GBR membrane materials is difficult to perform [31,32], and, to facilitate this purpose, the multi-step collagen manufacturing process from animal sources need be applied in the same way, which includes the chemical treatments, physical treatments, additives, molding, stabilization, drying and packaging/sterilization [26]. The in vitro tests are more proper than the animal in vivo tests from the viewpoint of time and cost consuming. For future development of collagenous biomaterials, the duplex tests of collagenase dissolution tests and TG-TDA might be useful.

The collagenous biomaterials appear to have advantage as the regenerative materials because these materials accelerate wound healing in soft tissues and lead to bone formation at osseous sites [1,2]. Biodegradable membranes in dentistry are now either collagen-based or polylactide based polymers. Both are preferred to non-biodegradable membranes (e.g., polytetrafluorethylene and titanium) because of the lack of need of secondary surgery. It should be cautioned, however, that biodegradable membranes show lower mechanical strength and are less efficient at space-making than non-biodegradable membranes. The clinical application of biodegradable membranes in dentistry is now limited to areas that are simple to set. Polylactide based polymers consist of polylactic acid (PLA), polyglycolic acid (PGA), polycaprolactone (PCL) and their copolymers (e.g., poly (lactic-co-glycolic acid) (PLGA) and poly (lactide-co-caprolactone) (PLCL)) [57,58,59]. These aliphatic polyester membranes can be industrially made with same quality, and are not influenced by the prion and virus contaminations [60]. The advantages of these membranes include their adjustable biodegradability and mechanical properties, which can be controlled by regulating the polymer composition. The addition of PCL increases the strength and degradability of PLA polymer structure, while PGA decreases these properties [56]. Although PLA- and PLGA-based membranes are non-cytotoxic and biodegradable, the release of oligomers and acid byproducts during degradation may often trigger inflammation reactions and a foreign body response in vivo because the degradation of polyester membranes is accomplished by hydrolysis [59]. On the other hand, collagen (especially, atelo-collagen) is enzymatically decomposed, causing a mild body reaction. The question of which material is more suitable for GBR membrane has not been answered yet, and both materials are simultaneously utilized in dentistry [2,18,59].

## 5. Conclusions

Within the limitations of this study using the dermis-type and membrane-type collagenous biomaterials which were characterized by four methods, the following conclusions were obtained:SEM observations confirmed that the dermis-type and membranous-type materials were fibrous and porous;TG-DTA revealed two characteristic DTG peak temperatures at about 220 °C and 320 °C. The combined specific DTG peak temperature appeared to imply the degree of cross-linking of collagen products;Collagenase dissolution tests directly indicated the level of cross-linking of collagen products, ranging from about 40 to 500 min;Subcutaneous implantation tests showed direct in vivo longevity of collagenous biomaterials; andThe membranous collagenous biomaterials were more cross-linked, leading to higher specific DTG peak temperature, larger collagenase dissolution time, and longer in vivo durability, compared with dermis-type ones because of intensified degree of cross-linking in collagen fibrils.

The numerical data obtained from the TG-DTA and collagenase dissolution tests might be newly indicative of the degree of cross-linking of the collagenous biomaterials, that is influential in determining the clinical performance of the collagenous biomaterials in terms of biodegradation and material life span. These data were well consistent with results of the subcutaneous implantation test in vivo. The set of the SEM observation, TG-DTA, collagenase dissolution test, and subcutaneous implantation test might offer a new method to assess the collagenous biomaterials for the material selection and development. It is expected to include mechanical tests such as tensile pulling in the future.

## Figures and Tables

**Figure 1 materials-15-01155-f001:**
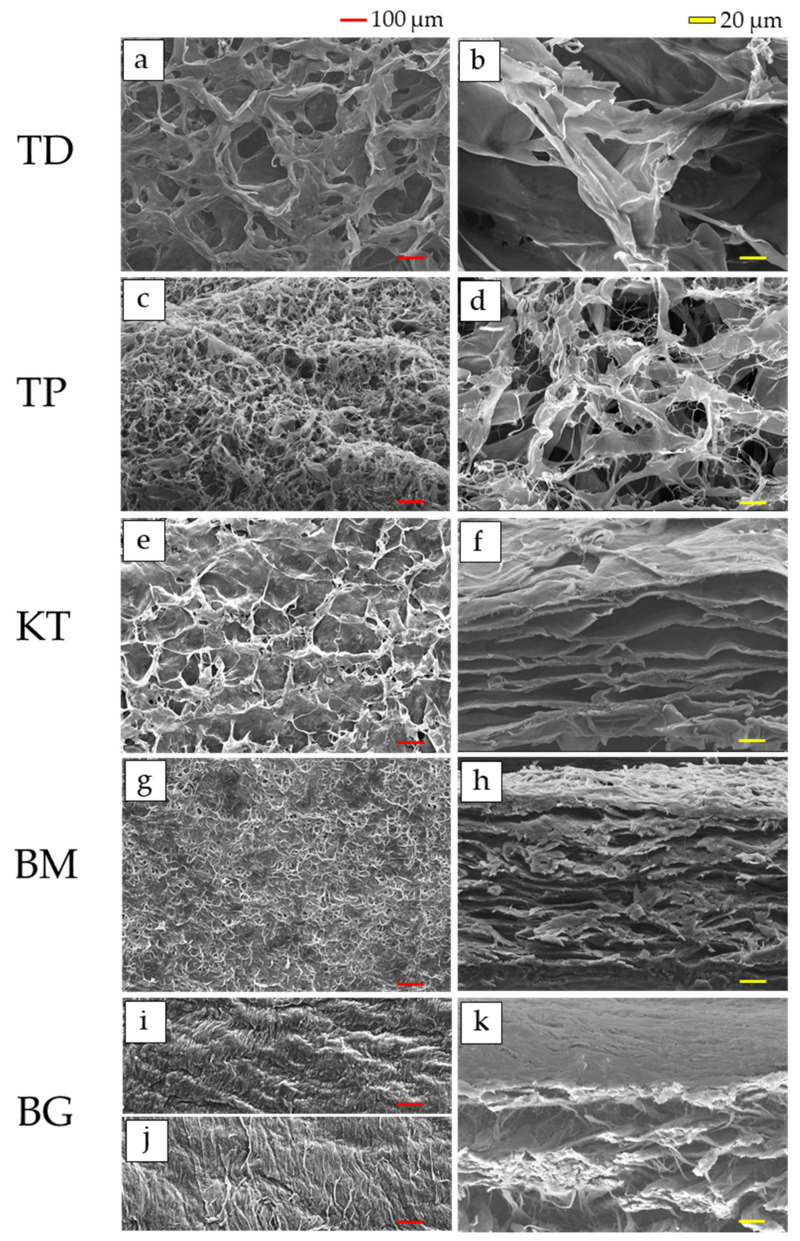
SEM micrographs of five collagenous biomaterials: (**a**,**b**) TD; (**c**,**d**) TP; (**e**,**f**) KT; (**g**,**h**) BM; and (**i**,**j**,**k**) BG (left: outer surface; 50× magnification, right: cross-sectional surface; 500× magnification). Abbreviations of each sample are shown in Table 1, respectively.

**Figure 2 materials-15-01155-f002:**
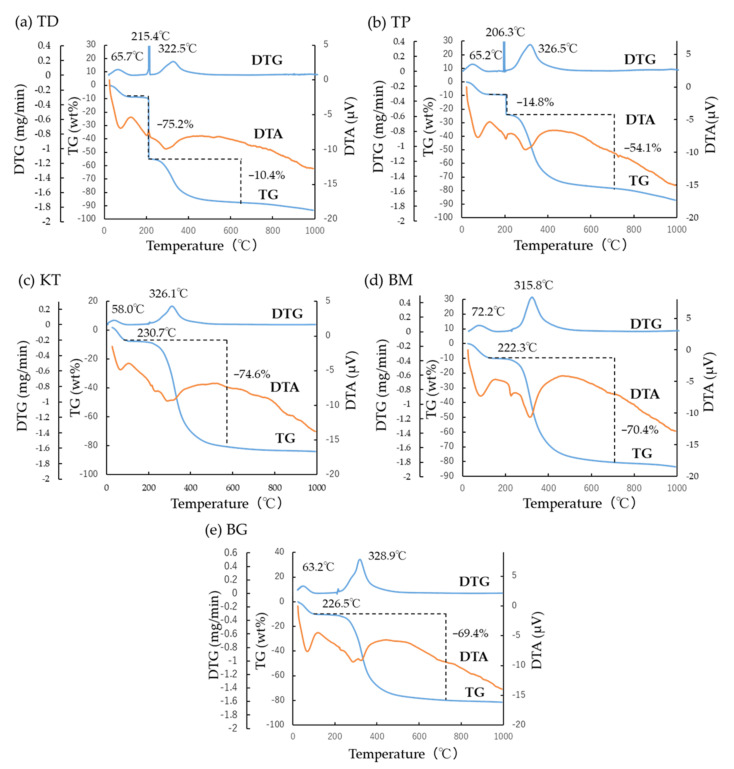
TG-DTA of five collagenous biomaterials: (**a**) TD; (**b**) TP; (**c**) KT; (**d**) BM; and (**e**) BG. Abbreviations of each sample are shown in Table 1, respectively.

**Figure 3 materials-15-01155-f003:**
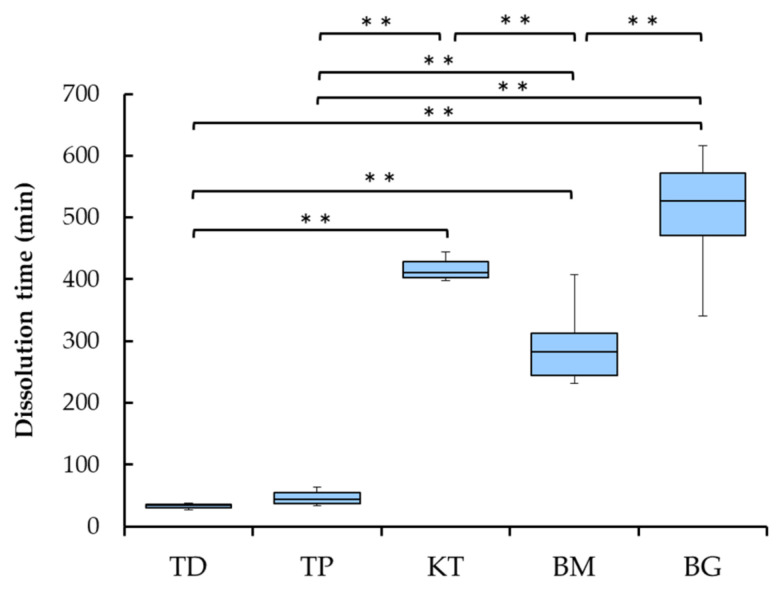
Box-plot of the collagenase dissolution tests of five collagenous biomaterials. Asterisk marks indicate significant differences (** *p* < 0.01). TD and TP were categorized as the dermis-type material and KT, BM, and BG were categorized as the membranous material, respectively. Abbreviations of each sample are shown in Table 1, respectively.

**Figure 4 materials-15-01155-f004:**
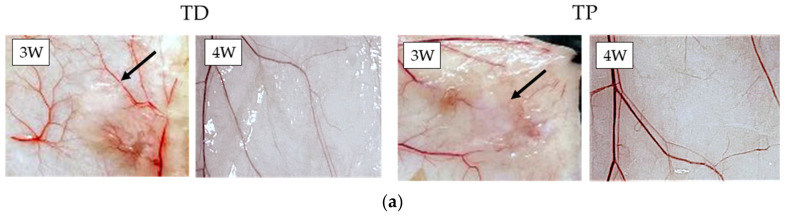
(**a**) Postoperative photographs of the dermis-type collagenous biomaterials (TD and TP) implanted in mouse subcutaneous tissues. These materials remained after three weeks (black arrows), but disappeared after four weeks. Abbreviations of each sample are shown in Table 1, respectively. (**b**) Postoperative photographs of the membranous collagenous biomaterials (KT, BM and BG) implanted in mouse subcutaneous tissues. These materials remained still nine weeks after implantation. Black arrows indicate the remained sample. Abbreviations of each sample are shown in Table 1, respectively.

**Figure 5 materials-15-01155-f005:**
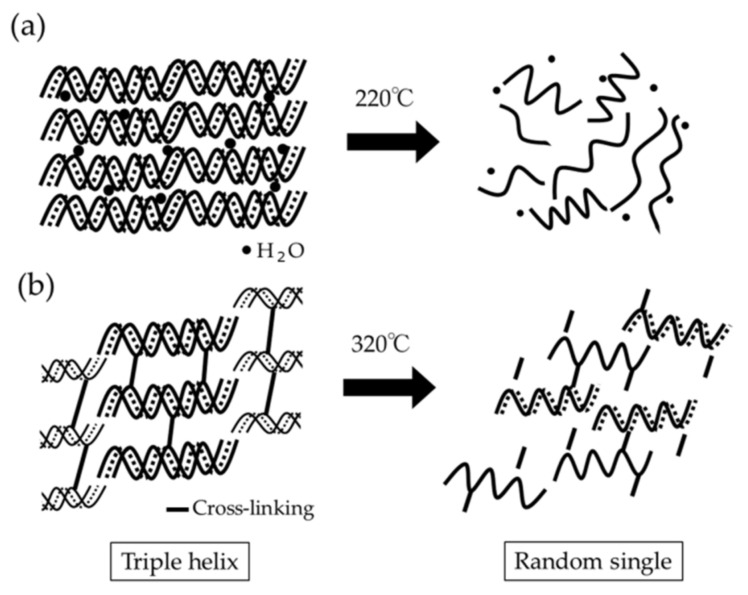
Schematics of structural changes of segments in collagenous biomaterials heated at 220 °C (**a**) and at 320 °C (**b**).

**Figure 6 materials-15-01155-f006:**
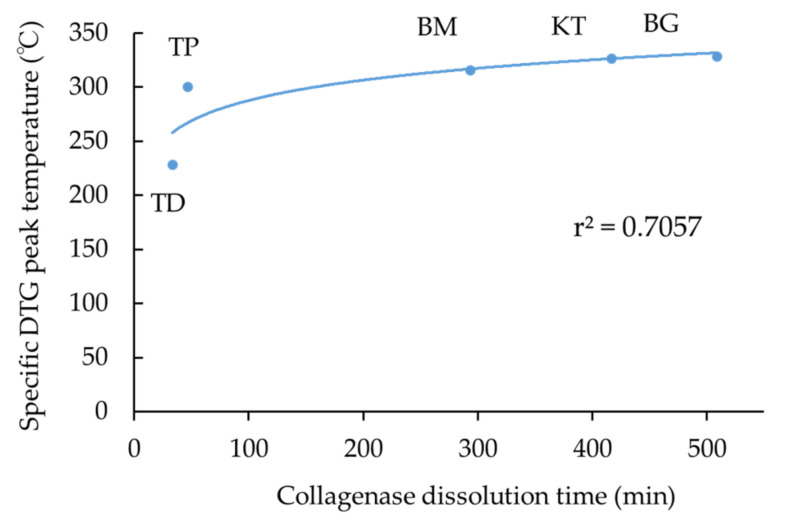
The logarithmic correlation between the specific DTG peak temperature and collagenase dissolution time of five collagenous biomaterials. Abbreviations of each sample are shown in Table 1, respectively.

**Table 1 materials-15-01155-t001:** Five collagenous biomaterials used in this study.

Product Name	Tissue Source (Cross-Linking)	Main Usage	Lot No.	Code
Terudermis	Bovine dermis (Yes)	Artificial dermis	M10063	TD
Teruplug	Bovine dermis (Yes)	Filling of tooth extraction socket	M1007F	TP
Koken Tissue Guide	Bovine dermis and tendon (Yes)	GTR membrane	19050A	KT
Biomend	Bovine tendon (Yes)	GTR/GBR membrane	1112022	BM
Bio-gide	Porcine dermis (No)	GTR/GBR membrane	81901547	BG

**Table 2 materials-15-01155-t002:** DTG peaks of five collagenous biomaterials, expressed in the temperature with the concurrent weight loss in the parenthesis.

Sample	Water Evaporation	Peak Temperature 1	Peak Temperature 2	Remained	Specific Peak Temperature
TD	65.7 °C	215.4 °C	322.5 °C		228.4 °C
	(7.0 wt%)	(75.2 wt%)	(10.4 wt%)	(7.4 wt%)	
TP	65.2 °C	206.3 °C	326.5 °C		300.7 °C
	(9.0 wt%)	(14.8 wt%)	(54.1 wt%)	(22.1 wt%)	
KT	58.0 °C	230.7 °C	326.5 °C		326.5 °C
	(7.5 wt%)	(virtually 0 wt%)	(74.6 wt%)	(17.9 wt%)	
BM	72.2 °C	222.3 °C	315.8 °C		315.8 °C
	(9.8 wt%)	(virtually 0 wt%)	(70.4 wt%)	(19.8 wt%)	
BG	63.2 °C	226.5 °C	328.9 °C		328.5 °C
	(9.7 wt%)	(0.3 wt%)	(69.1 wt%)	(20.9 wt%)	

Abbreviations of each sample are shown in Table 1, respectively.

## Data Availability

All data are included in the manuscript.

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
