# Peer review of "Characterization of Five Collagenous Biomaterials by SEM Observations, TG-DTA, Collagenase Dissolution Tests and Subcutaneous Implantation Tests"

_materials, 2022, doi:10.3390/ma15031155_

Round 1

Reviewer 1 Report

Recommendation: Minor revisions needed.

In this work, the authors characterized two dermis-type and three GTR/GBM membranous collagenous products by scanning electron microscopy (SEM), thermogravimetry-differential thermal analysis (TG-DTA), collagenase dissolution test, and subcutaneous implantation test in vivo. All of them are very interesting and valuable. In addition, the relative work is very systematic. Therefore, this manuscript should be considered for publication in materials, if the following questions can be addressed in the revision.

  • In the introduction section; some related collagenous products and variants which help to select the collagenous material for clinical use with the help of SEM, TG-DTA, … might be cited and discussed in detail. Moreover, it should be explained why these characterizations are helpful in selection steps.
  • Some extra characterizations would be helpful for the readers if it is possible for the authors to perform, such as checking the modulus of these materials or quantitative evaluation of the degree of cross-linking to give better ideas about mechanical properties of these materials, however, it is just optional to improve the quality of their valuable work.

Author Response

Response to the Reviewers' Comments (ID: materials-1553655):

-Reviewer 1-

・General comment:

We would like to thank the reviewer for careful and thorough reading of our manuscript. Corrected passages have been yellow highlighted in the whole of manuscript.

  1. In the introduction section; some related collagenous products and variants which help to select the collagenous material for clinical use with the help of SEM, TG-DTA, … might be cited and discussed in detail. Moreover, it should be explained why these characterizations are helpful in selection steps.

Answer: We have added the descriptions about some related collagenous products and variants using some references (No. 1-12) in the medical and dental fields, especially (Page 1, line 35- page 2, line 49). Also, we have added the descriptions about the characterization of the collagenous biomaterials with adding the references (No. 30-39, 43) why these characterizations are helpful in selection steps (Page 2, line 84- page 3, line 121).

        In addition, we have added the descriptions why we chose four methods used in this study; summary (Page 8, lines 256-259), SEM (Page 8, lines 260-261), TG-DTA (page 8, lines 274-276), and collagenase dissolution test (Page 9, lines 322-323). Thus, we have added the interpretation of our results and testing method in the discussion section (Page 11, lines 375-380) and clarify the main finding and the interpretation of this study in the conclusions section (Page 11, lines 422- page 12, line 429).

  1. Some extra characterizations would be helpful for the readers if it is possible for the authors to perform, such as checking the modulus of these materials or quantitative evaluation of the degree of cross-linking to give better ideas about mechanical properties of these materials, however, it is just optional to improve the quality of their valuable work.

Answer: Unfortunately, we did not clarify the mechanical properties and quantitative evaluation of the degree of cross-linking of the collagenous biomaterials in this study. However, we know the importance of these characteristic. So, we have added the explanations about the characterization of the collagenous biomaterials including these aspects (Page 2, line 84- page 3, line 121).

Reviewer 2 Report

This manuscript deals with the characterization of five collagenous biomaterials used in dentistry (i) one artificial dermis and one filling material for the tooth extraction socket, and (ii) three GTR/GBR membranes using SEM, TG-DTA, collagenase dissolution tests, and subcutaneous implantation tests. The figures are clear. The introduction is clear but should be improved. I can recommend the publication of this work after further changes. A few comments are added:

  • The introduction should be improved. Previous characterization studies of collagen biomaterials should be added and described in detail.
  • The production of collagen products is described very casually.
  • Misprint should be fixed: In 2.1 Materials Terudermis: TP should be replaced by Teruplug: TP
  • The authors should think carefully about the selection of the studied samples. Only one artificial dermis and only one filling material and 3 membrane materials, 1 of the membrane materials was a double-layered membrane material. It will be better to study at least 2 samples from each material. In addition, it will be useful to compare all of these characterizations on all the commercially available biomaterials.
  • It seems to me that TG-DTA analysis showed a peak around 220 °C for all the membrane materials, but the authors assigned this peak only to the BG sample. Can the authors comment on this assignment?
  • Can the authors comment on the effect of porosity on the collagenase dissolution test?
  • Can the authors comment on the effect of the chemical composition of cross-linking agents on the results of methods used?
  • Why did the authors not compare collagen biomaterials with another biomaterial (poly-lactide-polymer)?
  • I suggest concluding in sentences, not in points.
  • Can the authors comment and compare the results of their study (cross-linking etc.) and published article: Sheikh Z, Qureshi J, Alshahrani A. M., Nassar H., Ikeda Y., Glogauer M., Ganss B., Collagen based barrier membranes for periodontal guided bone regeneration applications, Odontology (2017) 105:1–12, DOI 10.1007/s10266-016-0267-0?
  • The authors should explain the main benefits of this study. The manufacturer produces studied materials for different use, therefore it is obvious that the materials will have different properties and different stability.

Author Response

Response to the Reviewers' Comments (ID: materials-1553655):

-Reviewer 2-

・General comment:

Thank you for your thorough review and your helpful advice and suggestions. We carefully considered your comments. Corrected passages have been yellow highlighted in the whole of manuscript.

  1. The introduction should be improved. Previous characterization studies of collagen biomaterials should be added and described in detail.

Answer: We have added the descriptions about some related collagenous products and variants using some references (No. 1-12), especially in the medical and dental fields (Page 1, line 35- page 2, line 49). Also, we have added the descriptions about the characterization of the collagenous biomaterials with adding previous studies (Reference No. 30-39, 43) why these characterizations are helpful in the clinical use (Page 2, line 84- page 3, line 121).

  1. The production of collagen products is described very casually.

Answer: Unfortunately, we could not obtain the manufacturing process of each materials because the manufacturers are not disclosed the original method. Thus, we have explained this part briefly.

(Page 2, lines 76-78)

Briefly, atelocollagen or purified type I collagen fibers were dispersed in an acidic media (pH 2.5), homogenized, filtered, reconstituted (pH 7), dehydrated, compressed, freeze dried, cross-linked (or not done), sized, packaged, and sterilized [28].

  1. Misprint should be fixed: In 2.1 Materials Terudermis: TP should be replaced by Teruplug: TP.

Answer: We have corrected the term from Terudermis: TP to Teruplug: TP (Page 3, line 131).

  1. The authors should think carefully about the selection of the studied samples. Only one artificial dermis and only one filling material and 3 membrane materials, 1 of the membrane materials was a double-layered membrane material. It will be better to study at least 2 samples from each material. In addition, it will be useful to compare all of these characterizations on all the commercially available biomaterials.

Answer: Thank you for your comments. We focused the collagenous biomaterials in this study. Some brands include the calcium phosphate, such as hydroxyapatite and octacalcium phosphate due to promote the bone formation. Therefore, we did not choose them. We also would like to choose three or more types of dermis-type and membrane, but we have listed up the commercially available collagenous products in our country to find at the preliminary research.

  1. It seems to me that TG-DTA analysis showed a peak around 220 °C for all the membrane materials, but the authors assigned this peak only to the BG sample. Can the authors comment on this assignment?

Answer: Thank you for pointing it out. We have added the respective degree at Peak temperature 1 of KT (230.7°C) and BM (222.3°C) in Figure 2 and Table 2 (Page 6, line 225 and Page 7, line 229). We have also rewritten the descriptions of the results for the membranous materials according to their changes (Page 6, lines 220-224).

  1. Can the authors comment on the effect of porosity on the collagenase dissolution test?

Answer: Thank you for pointing it out. We have added our interpretation of the collagenase dissolution test in the discussion section and explained how these data relate to SEM observations of the porosity of the collagenous biomaterials (Page 9, line 322- page 10, line 345).

  1. Can the authors comment on the effect of the chemical composition of cross-linking agents on the results of methods used?

Answer: Unfortunately, we could not answer because the manufacturers are not disclosed the method how these were processed (Please see our answer for you No.2 question as mentioned above).

However, we can describe and compare to the previous report (Please see your No.9 question below). It was not reported the reason about the effect of the chemical composition of cross-linking agents in this review article, but they indicated the difference in the resorption time depending on the presence or absence of cross-linking.

       (Page 10, lines 337-341)

Among the membranous materials, the non-cross-linked material (BG) showed significantly longevity in the dissolution test than the cross-linked materials (KT and BM). The obtained results in this in vitro study seemed to be consistent with the previous study [10], that the resorption time of the non-cross-linked material was longer than the cross-linked materials for clinical use.

  1. Why did the authors not compare collagen biomaterials with another biomaterial (poly-lactide-polymer)? I suggest concluding in sentences, not in points.

Answer: Thank you for your advice. We have added the descriptions of other materials in the introduction section (Page 2, lines 52-54) and the discussion section (Page 11, lines 381-403)

       Especially, we have added the descriptions of the comparison between the collagenous biomaterial and others in the discussion section.

  1. Can the authors comment and compare the results of their study (cross-linking etc.) and published article: Sheikh Z, Qureshi J, Alshahrani A. M., Nassar H., Ikeda Y., Glogauer M., Ganss B., Collagen based barrier membranes for periodontal guided bone regeneration applications, Odontology (2017) 105:1–12, DOI 10.1007/s10266-016-0267-0?

Answer: Thank you for your suggestion. We have added the descriptions of the comparison between our results and this reference [10] in the discussion section.

       (Page 10, lines 337-341)

Among the membranous materials, the non-cross-linked material (BG) showed significantly longevity in the dissolution test than the cross-linked materials (KT and BM). The obtained results in this study seemed to be consistent with the previous study [10], that the resorption time of the non-cross-linked material was longer than the cross-linked materials for clinical use.

  1. The authors should explain the main benefits of this study. The manufacturer produces studied materials for different use, therefore it is obvious that the materials will have different properties and different stability.

Answer: Thank you for pointing it out. Your points raised are quite right.

However, we hope that our selected methods will be applied not only for the evaluation of the commercial products but also newly developed materials. Thus, we have added descriptions why we chose four different methods and how it relates for clinical use in the discussion section (Page 8, lines 256-259, lines 260-261, lines 274-276; Page 9, line 322- page 10, line 345; and Page 11, lines 375-380).

Also, we have rewritten the descriptions in the conclusions section to clarify the main finding and the interpretation of this study (Page 11, line 422- page 12, line 429).

Reviewer 3 Report

Review on draft entitled “Characterization of Five Collagenous Biomaterials by SEM 2 Observations, TG-DTA, Collagenase Dissolution Tests and 3 Subcutaneous Implantation Tests”

In this research properties of collagenous biomaterials (two dermis-type and three membranous materials) were analyzed by four different methods: scanning electron microscopy (SEM), thermogravimetry-differential thermal analysis (TG-DTA), collagenase dissolution test, and subcutaneous implantation test in vivo.

The draft is interesting and rather well organized. It is in the scope of the journal, therefore it can be published after some improvements and corrections:

Questions and comments for the article:

Line 7:            please add the country to the affiliations.

Line 55:          the manuscript contains a few abbreviations and therefore the abbreviation list is not necessary.

Line 147:        the magnification in the description of the figure and above the figures is different. Please make it more clear which magnification is correct?

Line 197:        what is the size of the picture in figure 4? I see, that in pictures of TD and TP the blood vessels are very clear and bright. On the contrary, the blood vessels are almost invisible in pictures of KT, BM, and BG. So my question is if these pictures are of the same size and magnification? If the size is the same, why these blood vessels are so different?

Line 277:        figure 6 is doubtful. I don’t see any correlation. The trendline in the figure doesn’t have any sense. Please rearrange the figure.

Lines 300-3003: the last sentence of the discussion part should be extended to make it clearer.

The conclusions match the results, but it is still unclear which cross-linked or non-cross-linked collagenous biomaterials are more suitable for clinical use.

Author Response

Response to the Reviewers' Comments (ID: materials-1553655):

-Reviewer 3-

・General comment:

Thank you for your thorough review and your helpful comments. According to your kind suggestion, we have corrected the manuscript. Corrected passages have been yellow highlighted in the whole of manuscript.

・Questions and comments for the article:

  1. Line 7: please add the country to the affiliations.

Answer: We have added the term of the country to the affiliations (Page 1, line 8).

  1. Line 55: the manuscript contains a few abbreviations and therefore the abbreviation list is not necessary.

Answer: We have deleted the abbreviation ”UV” in the text (Page 2, lines 72-73) and the abbreviation list (Page 12, line 443).

  1. Line 147: the magnification in the description of the figure and above the figures is different. Please make it more clear which magnification is correct?

Answer: Thank you for pointing out our mistake. We have corrected the description of the magnification (left: 500× → 50×) in the figure caption. Also, we have added the respective scale bur in Figure 1 (Page 5, lines 200 and 201).

  1. Line 197: what is the size of the picture in figure 4? I see, that in pictures of TD and TP the blood vessels are very clear and bright. On the contrary, the blood vessels are almost invisible in pictures of KT, BM, and BG. So my question is if these pictures are of the same size and magnification? If the size is the same, why these blood vessels are so different?

Answer: According your suggestion, we have adjusted the pictures due to different tone of a color, brightness, and contrast because we have used different cameras among the samples (Page 8, line 250).

  1. Line 277: figure 6 is doubtful. I don’t see any correlation. The trendline in the figure doesn’t have any sense. Please rearrange the figure.

Answer: We have changed the trendline from the linear approximation to the logarithmic approximation in Figure 6 (Page 10, line 353). Also, we have corrected the description of the coefficient of determination (Page 8, lines 349-350).

  1. Lines 300-303: the last sentence of the discussion part should be extended to make it clearer.

Answer: Thank you for your advice. We have edited and rewritten this paragraph to clarify the material selection compared to other materials (Page 11, lines 381-403).

  1. The conclusions match the results, but it is still unclear which cross-linked or non-cross-linked collagenous biomaterials are more suitable for clinical use.

Answer: We have added the description of the summary in this section (Page 11, lines 417-420) and rewritten the final conclusion to clarify the main finding and the interpretation of this study (Page 11, line 422- page 12, line 429).

        Also, we have added the descriptions about the material selection in the discussion section which material is suitable for clinical use (That’s on a case-by-case choice) (Page 10, lines 342-345).

Reviewer 4 Report

The article presented by Hoshi, M. and Sawada, T. et. al. presents a research manuscript comparing five commercial collagenous biomaterials through morphology (SEM), level of crosslinking (TGA), durability (dissolution study), and in vivo longevity (subQ implantation). The authors suggest that the primary significance of this work presented herein is related to the selection of biomaterials for artificial dermis and guided bone regeneration in dentistry. However, the implications and characterization of commercially available collagen-based materials will also likely have implications toward mechanobiology applications and organoid cell culture generally as well.  The manuscript is generally well written, apart from a few statements where context, clarity, or citations could be added. Specifically, It is unclear to this reviewer whether dermis-type materials and membranous materials should be considered side-by-side. If these materials are used interchangeably or not, or how they are related should be commented on within the text. The Abstract, introduction, and concluding sections do not include summary findings and this information should be included within those sections. It is also difficult to assess the significance of these works as a majority of the studies (3/4) are not done quantitatively so the comparisons between these materials cannot be realistically made or placed among other collagen-based materials (or even amongst the selected materials). For these reasons, this reviewer cannot accept the manuscript in present form.  However, if these issues within the manuscript presented here are addressed, I believe that these works would provide much value. In future studies, the authors should aim to include female mice. Throughout history, many researchers have elected to only include male mice and this has resulted in poorer health outcomes for women.

Specific comments are below.

Citations should be added to statements on lines:

-50-52, “To be utilized in dental practice, extracted type I collagen and atelocollagen often need to be crosslinked to raise mechanical strength and enzymatic resistance.”’

 -53-54, “Chemical crosslinking is most achieved by glutaraldehyde,          carbodiimide/hydroxysuccinimide, diisocyanates, or oxidized sugar.”

Additional information should be provided in these instances:

-Lines 38-40, specify the organisms used for the in vivo studies.

-Lines 57-58, Provide same examples of collagenous products manufactured from type I collagen fibers derived from porcine skin, bovine skin, or bovine tendon in dentistry.

-Lines 61-62, clarify what is causing variations to take place and provide evidence for this in some form

-Table 1, It would be beneficial for the reader to have composition and manufacturing (processes) information in the table and manufacturer and Lot No. within the text rather than how it is organized.  

-Line 91, please clarify what surfaces are being torn.

-The number of replicates for experiments should be included within the manuscript

-Lines 119-122, “one-by-one” may imply that there were only individual mice per time point i.e. either 5 (one per time point) or 25 (one implant type each per time point). For either scenario there conclusions could not be determined with statistical significance.

-Figure 2. There are three plots per graph (TG, DTA, and DTG), however, DTA is not defined within the text.

The SEM observations, collagenase assay, and subcutaneous implantation tests seem to rely entirely on visual inspection without quantification. While visual descriptions of SEM images may be adequate, as pictures are included, quantification of image data should be performed through characterization of pore size/density and thickness of layering. This could be achieved using free software such as imageJ. Collagenase dissolution testing relying on visual inspection is inadequate, as there are many quantitative colormetric collagenase activity kit-assays available, and at the very least, snap-shot time-point images could be included as supplemental data to support the claims made here. The SubQ implantation test should include snap-shot images for each time point used. Measurements of implant size and perhaps mass remaining at each time points could have easily been recorded and plotted. If the physical samples are no longer available, the dimensions and changes in size could possibly be interrogated from photographs taken.  If observations were made at multiple time points these observations should be organized into a table. If images were taken at each time point they should all be included (perhaps only as supplemental, but nevertheless included).

Author Response

Response to the Reviewers' Comments (ID: materials-1553655):

-Reviewer 4-

・General comment:

Thank you for your thorough review and your helpful advice and suggestions. We carefully considered your comments. Corrected passages have been yellow highlighted in the whole of manuscript.

(1) However, the implications and characterization of commercially available collagen-based materials will also likely have implications toward mechanobiology applications and organoid cell culture generally as well.

Answer: Thank you for your suggestion. Unfortunately, we did not clarify the mechanobiology applications and organoid cell culture in this study. However, we know the importance of these characteristic. So, we have added the explanations about the characterization of the collagenous biomaterials including these aspects (Page 2, line 84- page 3, line 121).

(2) Specifically, It is unclear to this reviewer whether dermis-type materials and membranous materials should be considered side-by-side. If these materials are used interchangeably or not, or how they are related should be commented on within the text. The Abstract, introduction, and concluding sections do not include summary findings and this information should be included within those sections.

Answer: The points that reviewers suggested why was the comparison of the characterization between dermis-type and membranous materials can be found to be confuse. The first point is that collagenous biomaterials that are clinically applied in dentistry have a dermis -type and membranous type, both of which are materials for promoting bone formation. When the shape of materials supplied is different, the properties could be affected. Nevertheless, there are few reports to have been clarified on the point. In this study, we compared the properties of dermis-type and membranes made of collagen side by side.

And, we have rewritten and added the descriptions in this study to clarify our purpose and main findings in the Abstract (Page 1, lines 16-17 and lines 21-29), Introduction (Page 2, lines 78-83), and Discussion (Page 8, lines 256-259; page 9, line 322- page 10, line 345; page 11, lines 375-403) sections.

(3) It is also difficult to assess the significance of these works as a majority of the studies (3/4) are not done quantitatively so the comparisons between these materials cannot be realistically made or placed among other collagen-based materials (or even amongst the selected materials).

Answer: Thank you for your comments. We understand your point. However, how can you do a quantitative assessment in the SEM observations? Sure, we have measured at least 2 samples per one product. We have confirmed that there is no difference in the structural features between them. In the TG-DTA, we used the platinum container during the measurement. We firstly measured no sample with the container (blank) and checked the precision of the measurement. The sample could be maintain the temperature nearly constant during the reaction has made it possible to identify each stage in the TG-DTA. Thus, we could obtained the constant data and it is enough sample (n=1) for the commercial product measurement in this case. It should be check the data using some samples when we develop original samples. As you know, the animal tests are performed under the smallest number of animals for the animal welfare. Finally, we would like to make a decision for material selection while taking account of a comprehensive set of the results which were obtained by four methods.

(4) In future studies, the authors should aim to include female mice. Throughout history, many researchers have elected to only include male mice and this has resulted in poorer health outcomes for women.

Answer: Thank you for your advice. We will consider our future study using female mice if possible. So far, we avoided to use female mice because the female hormone is very influential for wound repair and bone formation. Also, we have to say it should be mentioned that the smallest number of animals are recommended to use for the animal welfare. Thank you for your understanding.

・Answer to the specific comments

Citations should be added to statements on lines:

  1. -50-52, “To be utilized in dental practice, extracted type I collagen and atelocollagen often need to be crosslinked to raise mechanical strength and enzymatic resistance.”’

Answer: We have added the reference [18] (Page 2, lines 69-70).

  1. -53-54, “Chemical crosslinking is most achieved by glutaraldehyde, carbodiimide/ hydroxysuccinimide, diisocyanates, or oxidized sugar.”

Answer: We have added the references [18,25,26] (Page 2, lines 70-72).

Additional information should be provided in these instances:

  1. -Lines 38-40, specify the organisms used for the in vivo studies.

Answer: We have added the term of the specific organisms (humans) in in vivo studies.

(page 2, lines 55-57)

the former sponge-like materials stay for at least 4 weeks in humans, while~

  1. Lines 57-58, Provide same examples of collagenous products manufactured from type I collagen fibers derived from porcine skin, bovine skin, or bovine tendon in dentistry.

Answer: We have added the description.

(page 2, lines 53-55)

         Gel and fiber scaffolds consisting of collagen have been applied for bone regeneration in clinical trials [1,2].

5.-Lines 61-62, clarify what is causing variations to take place and provide evidence for this in some form

Answer: Thank you for your advice. We know the variations of the manufacturing process from animal skins and tendons such as decellularization, acid solubilization, (not often, alkaline treatment), salt precipitate, filtration, dialysis, (sometimes, gel column separation, and electrophoresis) with repeated steps. However, these process are dependent on the manufacturers and are not disclosed by them. Thus, we did not describe the detail of the variations because detailing the manufacturing process is beyond the scope of this study.

  1. -Table 1, It would be beneficial for the reader to have composition and manufacturing (processes) information in the table and manufacturer and Lot No. within the text rather than how it is organized.

Answer: We have deleted the manufacturer name and added the tissue source and the presence or absence of cross-link of the collagenous biomaterials in Table 1 (page 3, line 143).

Product name

Tissue source (cross-link)

Main usage

Lot No.

Code

Terudermis

Bovine dermis (Yes)

Artificial dermis

M10063

TD

Teruplug

Bovine dermis (Yes)

Filling of tooth extraction socket

M1007F

TP

Koken Tissue Guide

Bovine dermis and tendon (Yes)

GTR membrane

19050A

KT

Biomend

Bovine tendon (Yes)

GTR/GBR membrane

1112022

BM

Bio-gide

Porcine dermis (No)

GTR/GBR membrane

81901547

BG

  1. -Line 91, please clarify what surfaces are being torn.

Answer: We have changed the explanation of this part to avoid any misunderstandings. The cross-section was exposed by cutting with scissors.

(page 4, line 146) Outer and cross-sectional surfaces of each sample

  1. -The number of replicates for experiments should be included within the manuscript

Answer: In the methods, we have added the sample number in SEM observation

(page 4, line 146) Outer and cross-sectional surfaces of each sample (n = 1)

And, to avoid any misunderstandings, we have changed the description of subcutaneous implantation tests.

(page 4, lines 174-176)

The mice were sacrificed at 1, 2, 3, 4 and 9 weeks after implantation, and the tissues surrounding the specimens were exposed and visually assessed using one mouse at each period.

  1. -Lines 119-122, “one-by-one” may imply that there were only individual mice per time point i.e. either 5 (one per time point) or 25 (one implant type each per time point). For either scenario there conclusions could not be determined with statistical significance.

Answer: We have already answered this questions about the sample number (Please see our answer for you No.8 question as mentioned above).

 (page 4, lines 174-176)

The mice were sacrificed at 1, 2, 3, 4 and 9 weeks after implantation, and the tissues surrounding the specimens were exposed and visually assessed using one mouse at each period.

And, we could not determine with the statistical significance in the implantation test because the animal tests are performed under the smallest number of animals for the animal welfare (Please see our answer for you No.(3) question as mentioned above).

  1. -Figure 2. There are three plots per graph (TG, DTA, and DTG), however, DTA is not defined within the text.

Answer: We have included the DTA and edited the description in the results of TG-DTA (Page 6, lines 210-224).

  1. The SEM observations, collagenase assay, and subcutaneous implantation tests seem to rely entirely on visual inspection without quantification. While visual descriptions of SEM images may be adequate, as pictures are included, quantification of image data should be performed through characterization of pore size/density and thickness of layering. This could be achieved using free software such as imageJ. Collagenase dissolution testing relying on visual inspection is inadequate, as there are many quantitative colormetric collagenase activity kit-assays available, and at the very least, snap-shot time-point images could be included as supplemental data to support the claims made here. The SubQ implantation test should include snap-shot images for each time point used. Measurements of implant size and perhaps mass remaining at each time points could have easily been recorded and plotted. If the physical samples are no longer available, the dimensions and changes in size could possibly be interrogated from photographs taken. If observations were made at multiple time points these observations should be organized into a table. If images were taken at each time point they should all be included (perhaps only as supplemental, but nevertheless included).

Answer: The aims of the collagenase dissolution test in this study is to investigate the fully absorption period of the materials. This in vitro test is conducted to predict it related to the clinical absorption period. We judge the absence or presence of the material by the visual inspection in the clinical treatment. That is one reason why we did not use the collagenase activity kit-assays and not evaluate the quantitative analysis. Another answer is that this method is for the quantitative analysis, but the degree of the cross linking is not detected because it also decomposed functional groups contained which are not involved in cross-linking in the material.

In addition, in the implantation test, we could not identify the implant area when the material is decomposed and absorbed. That is the reason why we could not use the snap-shot and not evaluate the quantitative analysis.

Round 2

Reviewer 2 Report

The authors provided all the suggested changes and answered all questions. Based on this fact, I can recommend the work for publication.

Author Response

We thank you for your review and accepting our paper.

Reviewer 4 Report

Many author responses to reviewer comments were good and adequately addressed concerns. Exceptions are listed below 

1R. However, the implications and characterization of commercially available collagen-based materials will also likely have implications toward mechanobiology applications and organoid cell culture generally as well.

1A: Thank you for your suggestion. Unfortunately, we did not clarify the mechanobiology applications and organoid cell culture in this study. However, we know the importance of these characteristic. So, we have added the explanations about the characterization of the collagenous biomaterials including these aspects (Page 2, line 84- page 3, line 121).

1R'. The response, within the text from Page 2, line 84- page 3, line 121 does not discuss any mechanobiology applications, but instead discusses a large variety of methods used for characterizing collagen-based materials. Of course, this may be useful information, but is tangential rather than directly related to the previous comment.

2R. Specifically, It is unclear to this reviewer whether dermis-type materials and membranous materials should be considered side-by-side. If these materials are used interchangeably or not, or how they are related should be commented on within the text. The Abstract, introduction, and concluding sections do not include summary findings and this information should be included within those sections.

2A. The points that reviewers suggested why was the comparison of the characterization between dermis-type and membranous materials can be found to be confuse. The first point is that collagenous biomaterials that are clinically applied in dentistry have a dermis -type and membranous type, both of which are materials for promoting bone formation. When the shape of materials supplied is different, the properties could be affected. Nevertheless, there are few reports to have been clarified on the point. In this study, we compared the properties of dermis-type and membranes made of collagen side by side.

And, we have rewritten and added the descriptions in this study to clarify our purpose and main findings in the Abstract (Page 1, lines 16-17 and lines 21-29), Introduction (Page 2, lines 78-83), and Discussion (Page 8, lines 256-259; page 9, line 322- page 10, line 345; page 11, lines 375-403) sections.

2R'. The responses within the text listed in this response do not clarify the question of interchangeability or justify side-by-side comparison  

3R. It is also difficult to assess the significance of these works as a majority of the studies (3/4) are not done quantitatively so the comparisons between these materials cannot be realistically made or placed among other collagen-based materials (or even amongst the selected materials).

3A. Thank you for your comments. We understand your point. However, how can you do a quantitative assessment in the SEM observations? Sure, we have measured at least 2 samples per one product. We have confirmed that there is no difference in the structural features between them. In the TG-DTA, we used the platinum container during the measurement. We firstly measured no sample with the container (blank) and checked the precision of the measurement. The sample could be maintain the temperature nearly constant during the reaction has made it possible to identify each stage in the TG-DTA. Thus, we could obtained the constant data and it is enough sample (n=1) for the commercial product measurement in this case. It should be check the data using some samples when we develop original samples. As you know, the animal tests are performed under the smallest number of animals for the animal welfare. Finally, we would like to make a decision for material selection while taking account of a comprehensive set of the results which were obtained by four methods.

3R'. SEM can be done quantitatively by assessing feature dimensions within the images. See Previous comment of, “quantification of image data should be performed through characterization of pore size/density and thickness of layering. This could be achieved using free software such as imageJ.” However, there is a variety of free software to do this with aside from ImageJ. TG-DTA was done quantitatively, however, with only one sample, it is impossible to make any meaningful statements. Sample number should be included in the text. See Responses to 4R' and 9R'

4R. In future studies, the authors should aim to include female mice. Throughout history, many researchers have elected to only include male mice and this has resulted in poorer health outcomes for women.

4A. Thank you for your advice. We will consider our future study using female mice if possible. So far, we avoided to use female mice because the female hormone is very influential for wound repair and bone formation. Also, we have to say it should be mentioned that the smallest number of animals are recommended to use for the animal welfare. Thank you for your understanding.

4R'. If female hormone is influential for wound repair, then it is essential that a cohort of female mice are included in this study. While using the smallest number of animals is always recommended, using so few animals as to lack statistical power is more wasteful, as the study and loss of life is totally without meaning.

5R.-Lines 61-62, clarify what is causing variations to take place and provide evidence for this in some form

5A. Thank you for your advice. We know the variations of the manufacturing process from animal skins and tendons such as decellularization, acid solubilization, (not often, alkaline treatment), salt precipitate, filtration, dialysis, (sometimes, gel column separation, and electrophoresis) with repeated steps. However, these process are dependent on the manufacturers and are not disclosed by them. Thus, we did not describe the detail of the variations because detailing the manufacturing process is beyond the scope of this study.

5R'. The Authors have not indicated a manuscript amendment in response to this comment

6R. -Line 91, please clarify what surfaces are being torn.

6A.  We have changed the explanation of this part to avoid any misunderstandings. The cross-section was exposed by cutting with scissors.

(page 4, line 146) Outer and cross-sectional surfaces of each sample

6R'. The Authors should clarify within the manuscript text, not simply in the responses to this reviewer, how samples are prepared regarding SEM

7R.-The number of replicates for experiments should be included within the manuscript

7A.  In the methods, we have added the sample number in SEM observation

(page 4, line 146) Outer and cross-sectional surfaces of each sample (n = 1)

7R'. The number of replicates for ALL experiments (not only SEM) should be included within the manuscript.  

8R. Figure 2. There are three plots per graph (TG, DTA, and DTG), however, DTA is not defined within the text.

8A.  We have included the DTA and edited the description in the results of TG-DTA (Page 6, lines 210-224).

8R'. I am unable to find the definition of the acronym, “DTA” within that section of text or anywhere else within the manuscript

9R. The SEM observations, collagenase assay, and subcutaneous implantation tests seem to rely entirely on visual inspection without quantification. While visual descriptions of SEM images may be adequate, as pictures are included, quantification of image data should be performed through characterization of pore size/density and thickness of layering. This could be achieved using free software such as imageJ. Collagenase dissolution testing relying on visual inspection is inadequate, as there are many quantitative colormetric collagenase activity kit-assays available, and at the very least, snap-shot time-point images could be included as supplemental data to support the claims made here. The SubQ implantation test should include snap-shot images for each time point used. Measurements of implant size and perhaps mass remaining at each time points could have easily been recorded and plotted. If the physical samples are no longer available, the dimensions and changes in size could possibly be interrogated from photographs taken. If observations were made at multiple time points these observations should be organized into a table. If images were taken at each time point they should all be included (perhaps only as supplemental, but nevertheless included).

9A. The aims of the collagenase dissolution test in this study is to investigate the fully absorption period of the materials. This in vitro test is conducted to predict it related to the clinical absorption period. We judge the absence or presence of the material by the visual inspection in the clinical treatment. That is one reason why we did not use the collagenase activity kit-assays and not evaluate the quantitative analysis. Another answer is that this method is for the quantitative analysis, but the degree of the cross linking is not detected because it also decomposed functional groups contained which are not involved in cross-linking in the material.

In addition, in the implantation test, we could not identify the implant area when the material is decomposed and absorbed. That is the reason why we could not use the snap-shot and not evaluate the quantitative analysis.

9R'. The Authors did not adequately respond to concerns herein, nor amend the manuscript with their response. There was no response to SEM concerns. While the response to Dissolution study is adequate, this rationale should be included in the manuscript. If there is no quantitative data regarding the dissolution study (i.e. collagenase kit assay) then, the methods should be amended to include the frequency of observation. If the Authors did not mark the area where the implant was placed, and in turn cannot find the area of skin after implant dissolution, then the snap-shots of time points where the implant IS still present should be included

10R. There are many grammatical errors within the amended text, See Lines

  • 82-84
  • 89-91
  • 110-111
  • 220-222
  • 222-224
  • 257-259
  • 274-277
  • 327-328
  • 330-332
  • 335-337
  • 342-345
  • 375-376
  • 376-377
  • 377
  • 381-383
  • 391-393
  • 422-426
  • 426-428

11R. Within the Amended Text, the authors have added the term GBR membranes (line 52, Page 2). “GBR” is not defined at this first instance of its mention but rather, defined later in the text.

Author Response

Response to the Reviewers' Comments (ID: materials-1553655) Round 2:

-Reviewer 4-

Thank you very much for providing important comments. We are thankful for the time and energy you expended.

We have addressed your comments with point-by-point responses, and revised the manuscript accordingly. Corrected passages have been yellow highlighted in the whole of manuscript. Our responses to the referees’ comments are as follow:

1R. However, the implications and characterization of commercially available collagen-based materials will also likely have implications toward mechanobiology applications and organoid cell culture generally as well.

1R'. The response, within the text from Page 2, line 84- page 3, line 121 does not discuss any mechanobiology applications, but instead discusses a large variety of methods used for characterizing collagen-based materials. Of course, this may be useful information, but is tangential rather than directly related to the previous comment.

Answer:

 We have checked the forth paragraphs in the Introduction section and added the descriptions of the mechanobiology applications (Page 3, Lines 133-136). We have also added the reference [44] (Page 16, Lines 598-599).

2R. Specifically, It is unclear to this reviewer whether dermis-type materials and membranous materials should be considered side-by-side. If these materials are used interchangeably or not, or how they are related should be commented on within the text. The Abstract, introduction, and concluding sections do not include summary findings and this information should be included within those sections.

2R'. The responses within the text listed in this response do not clarify the question of interchangeability or justify side-by-side comparison.

Answer:

We have checked and rewritten the question of interchangeability or justify side-by-side comparison in the Abstract (Page 1, Lines 16-19), Introduction (Page 2, Lines 53-55), and Discussion (Page 12, Lines 420-432) sections.

3R. It is also difficult to assess the significance of these works as a majority of the studies (3/4) are not done quantitatively so the comparisons between these materials cannot be realistically made or placed among other collagen-based materials (or even amongst the selected materials).

3R'. SEM can be done quantitatively by assessing feature dimensions within the images. See Previous comment of, “quantification of image data should be performed through characterization of pore size/density and thickness of layering. This could be achieved using free software such as imageJ.” However, there is a variety of free software to do this with aside from ImageJ. TG-DTA was done quantitatively, however, with only one sample, it is impossible to make any meaningful statements. Sample number should be included in the text. See Responses to 4R' and 9R'

Answer:

According to your suggestions, we have calculated the pore and layer sizes of the materials using the ImageJ software by the SEM observation. And, we have added the descriptions of these methods and results in the respective section.

Methods; Page 4, Lines 154-167

Results; Page 5, Lines 206-208, 212-214, 217-220, 223, 226-229

We have edited and rewritten the sample number in each experiment (Page 4, lines 167, 169, and 178; Page 5, line 194).

4R. In future studies, the authors should aim to include female mice. Throughout history, many researchers have elected to only include male mice and this has resulted in poorer health outcomes for women.

4A. Thank you for your advice. We will consider our future study using female mice if possible. So far, we avoided to use female mice because the female hormone is very influential for wound repair and bone formation. Also, we have to say it should be mentioned that the smallest number of animals are recommended to use for the animal welfare. Thank you for your understanding.

4R'. If female hormone is influential for wound repair, then it is essential that a cohort of female mice are included in this study. While using the smallest number of animals is always recommended, using so few animals as to lack statistical power is more wasteful, as the study and loss of life is totally without meaning.

Answer:

Exactly, the authors agree with the reviewer’s opinion. First of all, we apologize for the insufficient of words in a previous response of ours. In the literature, female mice have been selected when elder-model mice such as osteoporosis or postmenopausal and so on were performed. In this study, we assume that degradation of specimens is metabolism in the body, so we used young male mice with active metabolism in middle age at first. Considering the usage of dermis-type and membranous materials made of collagen, they should not be dissolved rapidly and disappears in body, so we needed to clarify at least the residual period due to metabolism. Also, as the reviewer pointed out, we recognize and agree with necessary to confirm the female mice with low metabolism or elders, assuming the actual effect in treatment. For the viewpoint of animal welfare, we stated not to enough that the minimum number that we wrote previously for considering subcutaneous tissue reaction and/or bone tissue reaction. Although we stated that a minimum number was recommended from the viewpoint of animal welfare in last response of ours, we did not have enough words in description. Following the reviewer's suggestions, we would like to proceed with the investigation of subcutaneous and bone tissue reactions using a minimum of female and male mice for a future work.

We have added the description about female mice and the influence of the female hormones (Page 4, Lines 195-196). We have also added the reference [46] (Page 16, Lines 601-602).

5R.-Lines 61-62, clarify what is causing variations to take place and provide evidence for this in some form

5A. Thank you for your advice. We know the variations of the manufacturing process from animal skins and tendons such as decellularization, acid solubilization, (not often, alkaline treatment), salt precipitate, filtration, dialysis, (sometimes, gel column separation, and electrophoresis) with repeated steps. However, these process are dependent on the manufacturers and are not disclosed by them. Thus, we did not describe the detail of the variations because detailing the manufacturing process is beyond the scope of this study.

5R'. The Authors have not indicated a manuscript amendment in response to this comment

Answer:

  According to your suggestion, we have indicated our previous comments in the text (Page 2, Lines 83-98).

6R. -Line 91, please clarify what surfaces are being torn.

6A. We have changed the explanation of this part to avoid any misunderstandings. The cross-section was exposed by cutting with scissors.

6R'. The Authors should clarify within the manuscript text, not simply in the responses to this reviewer, how samples are prepared regarding SEM

Answer:

We have added the description how to prepare the samples in the text in the M&M section (Page 4, lines 163-164).

”The cross-sectional surface of the sample was exposed by cutting with scissors.”

7R.-The number of replicates for experiments should be included within the manuscript

7R'. The number of replicates for ALL experiments (not only SEM) should be included within the manuscript.

Answer:

  We have edited and rewritten the sample number in each experiment (Page 4, lines 167, 169, and 178; Page 5, line 194).

8R. Figure 2. There are three plots per graph (TG, DTA, and DTG), however, DTA is not defined within the text.

8R'. I am unable to find the definition of the acronym, “DTA” within that section of text or anywhere else within the manuscript

Answer:

  We have added description about the definition of “DTA” in the M&M section (Page 4, Lines 175-176).

9R. While visual descriptions of SEM images may be adequate, as pictures are included, quantification of image data should be performed through characterization of pore size/density and thickness of layering. This could be achieved using free software such as imageJ.

Collagenase dissolution testing relying on visual inspection is inadequate, as there are many quantitative colormetric collagenase activity kit-assays available, and at the very least, snap-shot time-point images could be included as supplemental data to support the claims made here.

9R'. The Authors did not adequately respond to concerns herein, nor amend the manuscript with their response. There was no response to SEM concerns. While the response to dissolution study is adequate, this rationale should be included in the manuscript. If there is no quantitative data regarding the dissolution study (i.e. collagenase kit assay) then, the methods should be amended to include the frequency of observation. If the Authors did not mark the area where the implant was placed, and in turn cannot find the area of skin after implant dissolution, then the snap-shots of time points where the implant is still present should be included

Answer:

According to your suggestions, we have calculated the pore and layer sizes of the materials using the ImageJ software by the SEM observation. And, we have added the descriptions of these methods and results in the respective section. Please see the answer for 3R’.

Methods; Page 4, Lines 154-167

Results; Page 5, Lines 206-208, 212-214, 217-220, 223, 226-229

 About the dissolution test, we have added the descriptions of our visual inspection method in the M&M section (Page 4, Line 81) and the reason why we choice the visual inspection instead of the collagenase quantity measuring by the collagenase kit assay and  colorimetric assay (Page 11, Line 258-265).

 About the implantation test, we have added and edited the Figure 4a,b to show the comparison of the presence and absence the remained samples after the test (Page 9, Lines 280 and 284).

10R. There are many grammatical errors within the amended text, See Lines

82-84, 89-91, 110-111, 220-222, 222-224, 257-259, 274-277, 327-328, 330-332, 335-337, 342-345, 375-376, 376-377, 377, 381-383, 391-393, 422-426, 426-428

Answer:

 Thank you for your point our mistakes out. We have rechecked and edited the grammatical errors as listed below,

    (Page 2) Lines 92-94, (Page 3) Lines 99-101, 120-121,

    (Page 7) Lines 250-252, Lines 252-254, (Page 9) 291-293, 308-310,

    (Page 11) Lines 368-369, 374-379, 377-379, 384-387,

(Page 12) Lines 417-418, 418-419, 419-420,

    (Page 13) Lines 436-438, 447-448, 477-481, and 481-485.

11R. Within the Amended Text, the authors have added the term GBR membranes (line 52, Page 2). “GBR” is not defined at this first instance of its mention but rather, defined later in the text.

Answer:

We have already described the term and abbreviation of “GBR” in same line (Page 2, Line 53).
